# ASTGI: Adaptive Spatio-Temporal Graph Interactions for Irregular Multivariate Time Series Forecasting

**Xvyuan Liu**[1]\*, **Xiangfei Qiu**[1]\*, **Hanyin Cheng**[1], **Xingjian Wu**[1],
**Chenjuan Guo**[1], **Bin Yang**[1], **Jilin Hu**[1,2]✉

[1]School of Data Science and Engineering, East China Normal University, Shanghai, China
[2]Engineering Research Center of Blockchain Data Management, Ministry of Education, China
`{xvyuanliu, xfqiu, hycheng, xjwu}@stu.ecnu.edu.cn`
`{cjguo, byang, jlhu}@dase.ecnu.edu.cn`

## Abstract

Irregular multivariate time series (IMTS) are prevalent in critical domains like healthcare and finance, where accurate forecasting is vital for proactive decision-making. However, the asynchronous sampling and irregular intervals inherent to IMTS pose two core challenges for existing methods: (1) how to accurately represent the raw information of irregular time series without introducing data distortion, and (2) how to effectively capture the complex dynamic dependencies between observation points. To address these challenges, we propose the Adaptive Spatio-Temporal Graph Interaction (ASTGI) framework. Specifically, the framework first employs a Spatio-Temporal Point Representation module to encode each discrete observation as a point within a learnable spatio-temporal embedding space. Second, a Neighborhood-Adaptive Graph Construction module adaptively builds a causal graph for each point in the embedding space via nearest neighbor search. Subsequently, a Spatio-Temporal Dynamic Propagation module iteratively updates information on these adaptive causal graphs by generating messages and computing interaction weights based on the relative spatio-temporal positions between points. Finally, a Query Point-based Prediction module generates the final forecast by aggregating neighborhood information for a new query point and performing forecasting. Extensive experiments on multiple benchmark datasets demonstrate that ASTGI outperforms various state-of-the-art methods.

## 1 Introduction

Irregular Multivariate Time Series Forecasting (IMTSF) is a core problem across numerous critical scientific and engineering domains. Its applications are wide-ranging, from monitoring vital signs in intensive care units to tracking the evolution of environmental indicators in climate science (Yao et al., 2018; Vio et al., 2013; Shukla & Marlin, 2020; Brouwer et al., 2019; Bilos et al., 2021; Wang et al., 2023; Shao et al., 2025; Wu et al., 2025c; Cheng et al., 2026; Lu et al., 2011; Yang et al., 2023). This type of data is fundamentally defined by two characteristics: 1) intra-series irregularity, where observations of the same variable occur at unequal time intervals, and 2) inter-series asynchrony, where observation timestamps are misaligned across different variables. These traits make it challenging to directly apply existing models designed for regular time series. Furthermore, despite existing research dedicated to addressing irregular time series prediction tasks (Liu et al., 2026; Qiu et al., 2025a; 2024; Li et al., 2025b; Qiu et al., 2025b), current methodological paradigms are commonly constrained by two core challenges, which impede their application potential in complex real-world scenarios.

**The first challenge is how to accurately represent the original information of an irregular time series without introducing information distortion.** An accurate representation must preserve the original sampling pattern, as this pattern itself contains key information about the system's dynam-

---
\*Equal Contribution

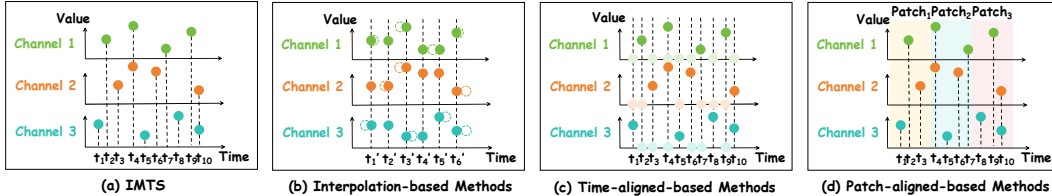

Figure 1: An illustration of information distortion in mainstream IMTSF paradigms. (a) Raw IMTS. (b) Interpolation-based: Converts irregular series into equally spaced series through numerical interpolation. Dashed circles denote artificially generated points that were not actually observed. (c) Time-aligned-based: Maps the observations of all variables to a unified timeline. Points on the zero line denote filled placeholders where no real observation exists for that variable. (d) Patch-aligned-based: Slices the time series into multiple patches.

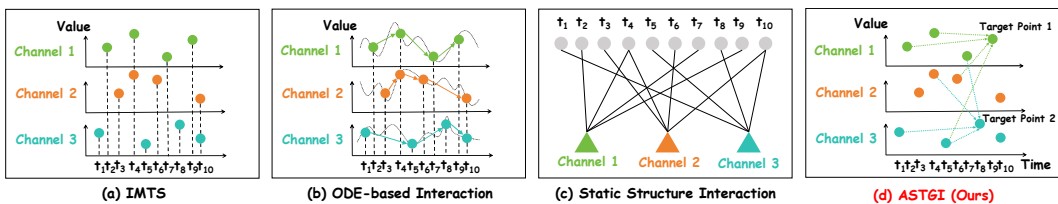

Figure 2: From fixed interaction rules to adaptive graph interaction. (a) Raw IMTS. (b) ODE-based interaction: Follows the temporal sequence to model continuous dynamics between observations. (c) Static Structure Interaction: Employs a fixed, predefined graph structure, confining information exchange to a static set of connections. (d) ASTGI (Ours): Adaptively constructs a unique graph for each observation point, enabling the capture of complex and dynamic dependencies.

ics (Shukla & Marlin, 2020; Li & Marlin, 2020). Conversely, any artificial alteration to the original data structure introduces information distortion, corrupting this intrinsic information and ultimately compromising the model's predictive performance (Ansari et al., 2023; Wang et al., 2026b;c; 2025c;b; Qiu et al., 2025c). There are three main representation paradigms, each with its own limitations. 1) Interpolation-based Methods (Figure 1b): This paradigm transforms an irregular series into an equally spaced series through numerical interpolation (Shukla & Marlin, 2021). However, this approach generates artificial data points that were not actually observed, which can introduce bias and distort the original sampling distribution (Shukla & Marlin, 2019; Zhang et al., 2023). 2) Time-aligned-based Methods (Figure 1c): This paradigm maps the observations of all variables to a unified timeline and fills in the missing values (Brouwer et al., 2019; Wang et al., 2025a; Yu et al., 2025b;a). Its main drawback is the loss of precise information about the time intervals between the original observations (Che et al., 2016; Rubanova et al., 2019). 3) Patch-aligned-based Methods (Figure 1d): This paradigm divides the time series into multiple patches (Cirstea et al., 2022; Nie et al., 2023; Wu et al., 2025a). However, a rigid division may disrupt the continuity of information. Furthermore, intra-patch aggregation can smooth out critical, fine-grained dynamics (Zhang et al., 2024b; Luo et al., 2025).

**The second challenge lies in how to effectively capture the complex dynamic dependencies among observation points.** It is important to note that these two challenges are closely coupled in a progressive manner: accurate representation serves as the prerequisite and foundation for effective dependency modeling. If the raw information is distorted during representation (Challenge 1), the subsequent modeling of dynamic dependencies (Challenge 2) will inevitably be built upon inaccurate data, fundamentally compromising the model's ability to capture true system dynamics. In many real-world scenarios, the interactions among points in an irregular time series are not static; they evolve dynamically over time. Accurately capturing these dependencies is crucial for understanding the system's behavior and making precise predictions (Chen et al., 2023b; Zhang et al., 2022; Qiu et al., 2025d;e; Wu et al., 2025b; 2026). However, existing methods generally rely on pre-defined and non-adaptive interaction structures. This means the scope of information exchange between observation points is often limited by a set of fixed, prior rules. For instance, in ODE-based Interaction (Figure 2b), information interaction strictly follows the temporal order, meaning an observation can only influence its immediate subsequent state on the timeline and cannot estab-

lish a direct connection with more distant historical observations (Rubanova et al., 2019; Brouwer et al., 2019; Schirmer et al., 2022; Bilos et al., 2021). Meanwhile, graph-based methods (Figure 2c) construct connections based on fixed rules, such as belonging to the same time point or the same variable (Yalavarthi et al., 2024; Li et al., 2025a; Chen et al., 2023a; Shang & Chen, 2024; Shang et al., 2024; 2026). The common limitation of these methods is their inability to dynamically and flexibly identify the truly relevant observation points based on the specific context of each point. As a result, they struggle to capture the deep dynamic correlations that span across time and variables.

To address the above challenges, we propose the Adaptive Spatio-Temporal Graph Interaction (ASTGI) framework (Figure 2d). This framework begins with a Spatio-Temporal Point Representation module that directly encodes each discrete time series observation into a point within a learnable spatio-temporal embedding space. This method operates on the original set of observation points without requiring interpolation or alignment, thus fully preserving the structure and patterns of the raw data and effectively avoiding information distortion, thereby addressing the first challenge. To tackle the second challenge, we then design a Neighborhood-Adaptive Graph Construction module, which adaptively builds a causal graph for each point in the embedding space via a nearest-neighbor search, replacing fixed a priori interaction rules. Subsequently, the Spatio-Temporal Dynamic Propagation module performs iterative information updates on these adaptive graphs, generating messages and calculating interaction weights based on the relative spatio-temporal positions between points. Finally, the framework's Query Point-based Prediction module yields the prediction by aggregating information from the neighborhood of a new query point and performing forecasting on it.

Our main contributions can be summarized as follows:

- To address IMTSF, we propose a general framework called ASTGI. It learns an accurate forecasting model through adaptive spatio-temporal graph interactions, which effectively avoids information distortion while flexibly capturing dynamic dependencies.

- We design the Neighborhood-Adaptive Graph Construction module, which discards pre-defined static interaction structures and adaptively constructs a causal interaction graph for each observation point by performing a nearest-neighbor search in the embedding space.

- We design a relation-aware dynamic propagation mechanism where message generation and interaction weighting are explicitly conditioned on the spatio-temporal relative positions between points, enabling the capture of highly context-dependent dynamics.

- We conduct extensive experiments on public datasets. The results show that ASTGI outperforms various state-of-the-art baselines. All datasets and code are available at `https://github.com/decisionintelligence/ASTGI`.

## 2  RELATED WORK

Deep learning has made impressive progress in natural language processing (NLP), time series analysis, computer vision, and other aspects Zhang et al. (2024a); Yu et al. (2025d;c;e); Zhang et al. (2025a); Chen et al. (2025); Lu et al. (2024a; 2023); Ma et al. (2024a;b); Zhou et al. (2025); Ma et al. (2025d); Wang et al. (2026a). Studies have shown that learned features may perform better than human-designed features Zhang et al. (2025c;b); Qi et al. (2025); Ma et al. (2025c); Lu et al. (2024b; 2025); Ma et al. (2025a;b). Existing deep learning methods for IMTSF can be broadly categorized into two paradigms based on how they handle data irregularity: Structured Representation-based methods and Raw-Data-based methods. The former transforms irregular data into regular structures to utilize standard sequence models, often at the cost of information distortion. The latter models discrete observations directly but typically relies on fixed rules for interaction. Our ASTGI framework falls into the second category but distinguishes itself by employing a fully adaptive graph interaction mechanism to capture dynamic dependencies without information loss.

### 2.1  STRUCTURED REPRESENTATION-BASED METHODS

The core idea of these methods is to convert irregular, asynchronous data into a regular format through structural transformation, thereby making it compatible with standard sequence models. This category includes the following mainstream approaches: (1) Interpolation-based Methods: This technique generates new values at missing time points through function fitting to create an equally

spaced time series, as adopted in works like mTAN (Shukla & Marlin, 2021). Its main limitation is the introduction of artificial data points that were not actually observed, which can alter the original data distribution and distort its intrinsic dynamic patterns (Shukla & Marlin, 2019; Zhang et al., 2023). (2) Time-Aligned-based Methods: This strategy maps the observations of all variables to a unified global timeline and fills in missing values, but this causes a loss of precise time interval information between original observations, thereby introducing distortion (Che et al., 2016; Rubanova et al., 2019; Brouwer et al., 2019). (3) Patch-Aligned-based Methods: To mitigate the sequence length problem caused by alignment, methods like t-PatchGNN (Zhang et al., 2024b) divide the timeline into fixed-size patches for local alignment. However, this aggregation based on a predefined granularity may smooth out or lose critical fine-grained dynamics within each patch (Luo et al., 2025; Nie et al., 2023). In contrast, our ASTGI framework directly represents each discrete observation, fully preserving the original data and fundamentally avoiding the information distortion caused by structural transformations.

## 2.2 Raw-Data-based Methods

Unlike the previous category, this paradigm models the set of discrete observation points directly, avoiding the distortion introduced by data structuring. However, it typically relies on predefined, non-adaptive rules to capture the dependencies between points. (1) ODE-based Interaction: Represented by models like Latent-ODE (Rubanova et al., 2019) and NeuralFlows (Bilos et al., 2021), these methods treat the evolution of a time series as a continuous dynamical system. Although they can naturally handle queries at any time point, their inherent Markov assumption strictly confines interactions to temporally adjacent states. This prevents the model from capturing direct long-range dependencies between non-adjacent events (Brouwer et al., 2019; Schirmer et al., 2022). (2) Static Structure Interaction: These methods use Graph Neural Networks (GNNs) to learn the relationships between observation points. However, their graph structure is typically constructed based on fixed heuristic rules (Yalavarthi et al., 2024; Li et al., 2025a). Such a static topology is insensitive to the specific data context and cannot adapt as the system state evolves, making it difficult to capture event-driven dynamic associations. Unlike the fixed interaction rules of these methods, ASTGI adaptively constructs an interaction graph for each observation point, enabling it to dynamically capture context-dependent dependencies.

## 3 Methodology

An IMTS sample can be formally represented as a set of discrete observations $\mathbb{S}$:

$$\mathbb{S} = \{(t_i, x_i, c_i)\}_{i=1}^N, \tag{1}$$

where $\mathbb{S}$ contains $N$ observation tuples. For each tuple $(t_i, x_i, c_i)$, $t_i \in \mathbb{R}$ is the timestamp of the observation, $x_i \in \mathbb{R}$ is the corresponding observed value, and $c_i \in \{1, \ldots, N_C\}$ is the variable index, indicating which of the $N_C$ variables the observation belongs to. This set-based representation naturally accommodates irregular sampling intervals and unaligned observations across variables.

For the forecasting task, given a split timestamp $t_s$, the sample $\mathbb{S}$ is partitioned into a historical set $\mathbb{S}_{\text{hist}}$ and a query set $\mathbb{S}_{\text{query}}$:

$$\mathbb{S}_{\text{hist}} = \{(t_i, x_i, c_i) \in \mathbb{S} \mid t_i \leq t_s\}, \mathbb{S}_{\text{query}} = \{(t_j, x_j, c_j) \in \mathbb{S} \mid t_j > t_s\}. \tag{2}$$

The goal is to learn a forecasting model $\mathcal{F}$. This model takes the historical observation set $\mathbb{S}_{\text{hist}}$ and a set of query coordinates $\mathbb{Q} = \{(t_j, c_j)\}$ as input, and predicts the corresponding set of true values $\mathbb{X}_q = \{x_j\}$. The entire forecasting process can be represented as:

$$\mathcal{F}(\mathbb{S}_{\text{hist}}, \mathbb{Q}) \rightarrow \hat{\mathbb{X}}_q, \tag{3}$$

where $\hat{\mathbb{X}}_q$ is the prediction for the true values $\mathbb{X}_q$.

## 3.1 Framework Overview

Figure 3 shows the adaptive Spatio-Temporal Graph Interaction (ASTGI) framework, which comprises four core stages: (a) Spatio-Temporal Point Representation, which directly encodes each

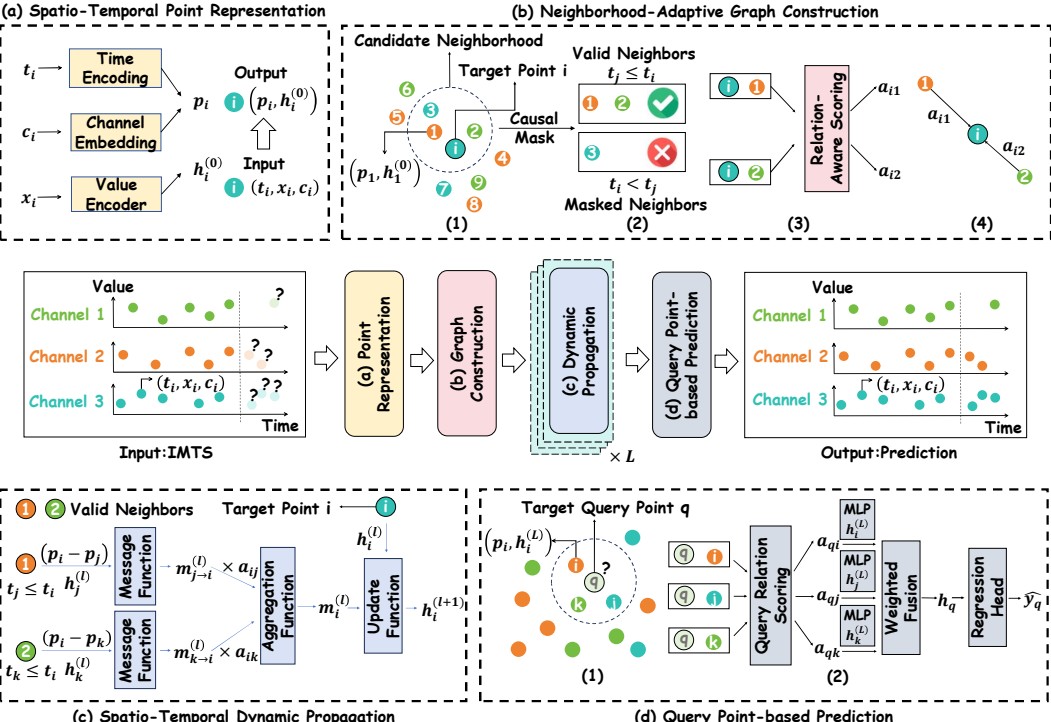

Figure 3: Overview of the ASTGI framework. (a) Directly representing each discrete observation as a spatio-temporal point. (b) Adaptively constructing a causal graph for each point. (c) Iteratively propagating information on the adaptive graphs to update features. (d) Unifying prediction as a neighborhood aggregation task for a query point.

discrete observation into a point in a spatio-temporal embedding space, preserving the integrity of the original data; (b) Neighborhood-Adaptive Graph Construction, which adaptively builds the graph structure and interaction weights based on the proximity of points in the embedding space; (c) Spatio-Temporal Dynamic Propagation, which updates point states through a multi-layer graph message passing mechanism to capture deep dependencies; and finally, (d) Query Point-based Prediction, which unifies the prediction task as a regression problem for a new query point's attributes within this space.

## 3.2 SPATIO-TEMPORAL POINT REPRESENTATION

The first step in our framework is to represent each discrete observation $(t_i, x_i, c_i)$ from the historical set $\mathbb{S}_{\text{hist}}$ as a structured spatio-temporal point. This transformation is achieved by introducing three dedicated encoders: (1) a Channel Embedding, which uses a learnable embedding matrix $\boldsymbol{E}_C \in \mathbb{R}^{N_C \times d_c}$ to map the variable index $c_i$ to an embedding vector $\boldsymbol{e}_{c_i}$, capturing the intrinsic relationships between different variables; (2) a Time Encoding, which employs a learnable Multi-Layer Perceptron (MLP) $\Phi_T : \mathbb{R} \to \mathbb{R}^{d_t}$ to map the timestamp $t_i$ to a time embedding $\boldsymbol{e}_{t_i}$, allowing it to flexibly learn complex temporal patterns; and (3) a Value Encoder, which uses another independent MLP $\Phi_X : \mathbb{R} \to \mathbb{R}^{d_{\text{model}}}$ to map the observation value $x_i$ into an initial $d_{\text{model}}$-dimensional feature vector $\boldsymbol{h}_i^{(0)}$. This vector will be iteratively updated in the subsequent dynamic propagation layers.

We concatenate channel embedding and time embedding to form the spatio-temporal coordinate $\boldsymbol{p}_i$ for each observation point, which defines its position in a learned $(d_c + d_t)$-dimensional space:

$$\boldsymbol{p}_i = \boldsymbol{e}_{c_i} \oplus \boldsymbol{e}_{t_i} \in \mathbb{R}^{d_c + d_t}, \tag{4}$$

where $\oplus$ denotes vector concatenation. With this, the original set of discrete observations is transformed into a set of spatio-temporal points $\{(\boldsymbol{p}_i, \boldsymbol{h}_i^{(0)}) \mid (t_i, x_i, c_i) \in \mathbb{S}_{\text{hist}}\}$. This representation preserves every original observation point in its entirety, thereby avoiding issues common to interpolation or alignment paradigms, such as information distortion and the introduction of artificial data

points. The core of this representation lies in the spatio-temporal coordinate space $\mathbb{R}^{d_c+d_t}$, formed by the channel and time embeddings, which directly serves as a embedding space. It is important to emphasize that the spatial dimension here does not refer to physical geographic location. Instead, it is an abstract dimension learned from the data, designed to capture the intrinsic relationships between different variables. The subsequent graph construction is then adaptively defined based entirely on the proximity of points within this learned space. Notably, the spatio-temporal coordinate $\boldsymbol{p}_i$ remains fixed throughout the subsequent propagation process, serving as a stable positional anchor, while only the feature vector $\boldsymbol{h}_i$ is iteratively refined.

### 3.3 NEIGHBORHOOD-ADAPTIVE GRAPH CONSTRUCTION

To overcome the limitations of predefined, static interaction structures, we do not presuppose a global graph structure. Instead, we adaptively construct a directed and weighted causal graph for each spatio-temporal point $\{(\boldsymbol{p}_i, \boldsymbol{h}_i^{(0)}) \mid (t_i, x_i, c_i) \in \mathbb{S}_{\text{hist}}\}$. For clarity, we will refer to an arbitrary spatio-temporal point as point $i$, and one of its neighboring points as point $j$.

#### 3.3.1 CANDIDATE NEIGHBORHOOD IDENTIFICATION

We screen for the most relevant interaction candidates for each point $i$ using a two-step process. First, we identify a candidate neighborhood $\mathcal{C}(i)$ by searching through all historical points in $\mathbb{S}_{\text{hist}}$. Specifically, we select the $K$ points that are closest to point $i$ in the learned spatio-temporal coordinate space, measured by the Euclidean distance $\|\boldsymbol{p}_i - \boldsymbol{p}_j\|_2$. Subsequently, to ensure that information flows only from the past to the future, we apply a Causal Mask to this candidate set, removing all points from $\mathcal{C}(i)$ with timestamps later than $t_i$ to obtain the final, valid set of neighbors $\mathcal{N}(i)$.

#### 3.3.2 RELATION-AWARE SCORING

The influence of a point $j$ on a point $i$ is quantified by a dynamically computed interaction weight $a_{ij}$. Since our information propagation is an iterative process over multiple layers, this weight is re-calculated at each propagation layer $l$. The core of this calculation is a Relation-Aware Scoring function. At the $l$-th propagation layer, we define a relation vector $\boldsymbol{r}_{ij}^{(l)}$ to comprehensively describe the adaptive relationship between two points:

$$\boldsymbol{r}_{ij}^{(l)} = (\boldsymbol{p}_i - \boldsymbol{p}_j) \oplus \boldsymbol{h}_i^{(l)} \oplus \boldsymbol{h}_j^{(l)} \in \mathbb{R}^{(d_c+d_t)+2d_{\text{model}}}. \tag{5}$$

This vector combines the relative position of the two points in the spatio-temporal coordinate space, $(\boldsymbol{p}_i - \boldsymbol{p}_j)$, with the current features of both interacting parties. This relation vector is fed into a small MLP network, $\text{MLP}_{\text{score}}$, to generate a raw interaction score $s_{ij} = \text{MLP}_{\text{score}}(\boldsymbol{r}_{ij}^{(l)})$. Finally, we apply the Softmax function to normalize the interaction scores over all valid neighbors (after causal masking) to obtain the final interaction weight $a_{ij}$:

$$a_{ij} = \frac{\exp(s_{ij})}{\sum_{k \in \mathcal{N}(i)} \exp(s_{ik})}. \tag{6}$$

### 3.4 SPATIO-TEMPORAL DYNAMIC PROPAGATION

On the adaptively constructed graphs, we stack $L$ information propagation layers to update the point features, thereby capturing long-range and complex spatio-temporal dependencies. At layer $l$, the feature $\boldsymbol{h}_i^{(l)}$ of each point $i$ is updated following a message-aggregation-update framework. While the graph topology $\mathcal{N}(i)$ remains fixed across layers, the interaction weights $a_{ij}$ are recomputed at each layer based on the updated features, allowing the model to progressively refine its attention over neighbors as representations evolve.

#### 3.4.1 MESSAGE FUNCTION

The first step in information propagation is to define the message passed from a neighbor node $j$ to node $i$. To achieve a relation-aware interaction, our message function depends not only on the sender's state $\boldsymbol{h}_j^{(l)}$ but also explicitly takes the spatio-temporal displacement vector $(\boldsymbol{p}_i - \boldsymbol{p}_j)$ as

input. This design allows the model to modulate the transmitted information based on the relative spatio-temporal position of the neighbor to the target point. The message $\boldsymbol{m}_{j \to i}^{(l)}$ is generated by a Multi-Layer Perceptron network, $\text{MLP}_{\text{msg}}$:

$$\boldsymbol{m}_{j \to i}^{(l)} = \text{MLP}_{\text{msg}}(\boldsymbol{h}_j^{(l)} \oplus (\boldsymbol{p}_i - \boldsymbol{p}_j)). \tag{7}$$

### 3.4.2 Aggregation Function

Next, node $i$ aggregates all incoming messages from its causal neighborhood $\mathcal{N}(i)$ through a weighted sum. The aggregation weights $a_{ij}$ are calculated via the dynamic scoring mechanism in Section 3.3, reflecting the relative importance of each neighbor in the current interaction. This aggregation operation can adaptively focus on the most informative neighbors:

$$\boldsymbol{m}_i^{(l)} = \sum_{j \in \mathcal{N}(i)} a_{ij} \cdot \boldsymbol{m}_{j \to i}^{(l)}. \tag{8}$$

### 3.4.3 Update Function

Finally, we use an update module with a residual connection and Layer Normalization to update point $i$'s feature, integrating its own historical information with the aggregated neighborhood information:

$$\boldsymbol{h}_i^{(l+1)} = \text{LayerNorm}(\boldsymbol{h}_i^{(l)} + \text{MLP}_{\text{update}}(\boldsymbol{m}_i^{(l)})). \tag{9}$$

After $L$ layers of propagation, we obtain the feature representation $\boldsymbol{h}_i^{(L)}$ for each observation point.

## 3.5 Query Point-based Prediction

We unify the prediction task into this framework. A prediction request for a target time $t_q$ and target variable $c_q$ is treated as a query point, and its value is predicted by applying regression to the aggregated information from its historical neighborhood. The key difference between this prediction process and the feature propagation stage lies in the network modules used. Instead of reusing the scoring and message networks from the feature propagation layers, we design a separate set of scoring and fusion networks specifically for the forecasting task. This parameter separation allows the model to optimize independently for two functionally distinct sub-tasks—iterative feature updating during multi-layer propagation and direct numerical regression at the final prediction step—thereby improving prediction flexibility and accuracy.

### 3.5.1 Query Point Embedding and Neighborhood Identification

We use the same encoders as for the historical points to map the query coordinates $(t_q, c_q)$ to a spatio-temporal position $\boldsymbol{p}_q = \boldsymbol{e}_{c_q} \oplus \Phi_T(t_q)$. Subsequently, we retrieve its $K$ nearest neighbors from all historical spatio-temporal points to form its neighborhood $\mathcal{N}(q)$. The causality is naturally satisfied as all historical points precede the query point in time.

### 3.5.2 Query Relation Scoring and Weighted Fusion

We use a query relation scoring network $\text{MLP}_{\text{query\_score}}$, dedicated to prediction, to compute the association score $s_{qi}$ between the query point $\boldsymbol{p}_q$ and each of its neighbors $i \in \mathcal{N}(q)$. This score depends on the spatio-temporal relative position between the two points and the neighbor's final feature state $\boldsymbol{h}_i^{(L)}$ after $L$ layers of propagation:

$$s_{qi} = \text{MLP}_{\text{query\_score}} \left( (\boldsymbol{p}_q - \boldsymbol{p}_i) \oplus \boldsymbol{h}_i^{(L)} \right). \tag{10}$$

These scores $s_{qi}$ are normalized through a Softmax function to obtain a set of interaction weights $a_{qi}$. Finally, these weights are used to perform a weighted fusion of neighbor information to generate a fusion vector $\boldsymbol{h}_q$. Before aggregation, we also use a value network $\text{MLP}_{\text{value}}$ to transform the neighbor features to extract the most valuable information segments for the prediction:

$$\boldsymbol{h}_q = \sum_{i \in \mathcal{N}(q)} a_{qi} \cdot \text{MLP}_{\text{value}}(\boldsymbol{h}_i^{(L)}). \tag{11}$$

Finally, this fusion vector $\boldsymbol{h}_q$, which aggregates spatio-temporal information from the neighborhood, is fed into a final Regression Head $\Phi_{\text{head}}$ to output the predicted value $\hat{x}_q = \Phi_{\text{head}}(\boldsymbol{h}_q)$.

Table 1: Statistical overview of the four benchmark datasets used in our experiments.

| Property | MIMIC | PhysioNet | Human Activity | USHCN |
|---|---|---|---|---|
| Domain | Healthcare | Healthcare | Biomechanics | Climate |
| # Samples | 21,250 | 11,981 | 1,359 | 1,114 |
| # Variables | 96 | 36 | 12 | 5 |
| Max Sequence Length | 96 | 47 | 131 | 337 |
| Avg # Observations | 144.6 | 308.6 | 362.2 | 313.5 |

## 3.6 TRAINING OBJECTIVE

The entire ASTGI model is end-to-end differentiable. During training, the model takes the history set $\mathbb{S}_{\text{hist}}$ as input to predict the value $x_j$ for each query coordinate $(t_j, c_j)$ from the query set $\mathbb{S}_{\text{query}}$. We jointly optimize all model parameters by minimizing the Mean Squared Error (MSE) loss function $\mathcal{L}$ over all queries:

$$\mathcal{L} = \frac{1}{|\mathbb{S}_{\text{query}}|} \sum_{(t_j, x_j, c_j) \in \mathbb{S}_{\text{query}}} (\hat{x}_j - x_j)^2, \tag{12}$$

where $\hat{x}_j$ is the prediction for the query $(t_j, c_j)$, and $x_j$ is its ground truth value.

## 4 EXPERIMENTS

### 4.1 EXPERIMENTAL SETUP

**Datasets and Baselines.** We conduct experiments on four widely-used public IMTS datasets spanning diverse application domains: MIMIC, PhysioNet, Human Activity, and USHCN. A statistical overview of these datasets is provided in Table 1. These four benchmarks collectively cover a broad spectrum of real-world irregularity characteristics, ranging from high-dimensional clinical records (MIMIC with 96 variables) to low-dimensional climate monitoring (USHCN with 5 variables), and from densely sampled motion signals (Human Activity) to sparsely observed physiological indicators (MIMIC). To ensure fairness and comparability of the results, all datasets are preprocessed by strictly following the standard procedures established in prior state-of-the-art works (Yalavarthi et al., 2024; Zhang et al., 2024b). We uniformly split the data into training, validation, and test sets with a ratio of 80%, 10%, and 10%, respectively. We selected a total of twelve state-of-the-art models from two main categories designed for irregular time series as baselines for comparison. For a detailed description of the datasets and baselines, please refer to the Appendix B.1 and B.2.

**Implementation Details.** All our experiments were conducted on a server equipped with an NVIDIA A800 GPU and implemented using the PyTorch 2.6.0+cu124 framework. All models are trained using the Mean Squared Error (MSE) as the loss function and optimized with the AdamW optimizer. We set the maximum number of training epochs to 300 and employ an early stopping strategy, where training is terminated if the model's performance on the validation set does not improve for 5 consecutive epochs. To ensure a fair comparison across all models, we primarily adopted the hyperparameter settings reported in the original papers for the baseline models. Building on these configurations, we conducted further search and fine-tuning of key hyperparameters on the validation set for some models to ensure that each achieved a competitive level of performance. To ensure reproducibility and mitigate the effects of randomness, each experiment is run independently with five different random seeds, and we report the mean and standard deviation. Detailed hyperparameter configurations for all models are provided in the Appendix B.2.

### 4.2 MAIN RESULTS

We present the performance comparison of ASTGI against the selected baselines on four public datasets—see Table 2. We have the following key observations: (1) ASTGI achieves state-of-the-art prediction accuracy across all datasets. Compared to the second-best performing model, Hi-Patch, ASTGI achieves significant reductions of approximately 6.04% in MSE. (2) ASTGI demonstrates consistent and superior performance across diverse domains. On datasets from healthcare (MIMIC,

Table 2: Forecasting performance on four IMTS datasets. Overall performance is evaluated by MSE and MAE (mean ± std). The best and second-best results are highlighted in **bold** and with an underline, respectively.

| Dataset | Human Activity | | USHCN | | PhysioNet | | MIMIC | |
|---|---|---|---|---|---|---|---|---|
| Metric | MSE | MAE | MSE | MAE | MSE | MAE | MSE | MAE |
| PrimeNet | 4.2507±0.0041 | 1.7018±0.0011 | 0.4930±0.0015 | 0.4954±0.0018 | 0.7953±0.0000 | 0.6859±0.0001 | 0.9073±0.0001 | 0.6614±0.0001 |
| NeuralFlows | 0.1722±0.0090 | 0.3150±0.0094 | 0.2087±0.0258 | 0.3157±0.0187 | 0.4056±0.0033 | 0.4466±0.0027 | 0.6085±0.0101 | 0.5306±0.0066 |
| CRU | 0.1387±0.0073 | 0.2607±0.0092 | 0.2168±0.0162 | 0.3180±0.0248 | 0.6179±0.0045 | 0.5778±0.0031 | 0.5895±0.0092 | 0.5151±0.0048 |
| mTAN | 0.0993±0.0026 | 0.2219±0.0047 | 0.5561±0.2020 | 0.5015±0.0968 | 0.3809±0.0043 | 0.4291±0.0035 | 0.9408±0.1126 | 0.6755±0.0459 |
| SeFT | 1.3786±0.0024 | 0.9762±0.0007 | 0.3345±0.0022 | 0.4083±0.0084 | 0.7721±0.0021 | 0.6760±0.0029 | 0.9230±0.0015 | 0.6628±0.0008 |
| GNeuralFlow | 0.3936±0.1585 | 0.4541±0.0841 | 0.2205±0.0421 | 0.3286±0.0412 | 0.8207±0.0310 | 0.6759±0.0100 | 0.8957±0.0209 | 0.6450±0.0072 |
| GRU-D | 0.1893±0.0627 | 0.3253±0.0485 | 0.2097±0.0493 | 0.3045±0.0305 | 0.3419±0.0029 | 0.3992±0.0011 | 0.4759±0.0100 | 0.4526±0.0055 |
| Raindrop | 0.0916±0.0072 | 0.2114±0.0072 | 0.2035±0.0336 | 0.3029±0.0264 | 0.3478±0.0019 | 0.4044±0.0020 | 0.6754±0.1829 | 0.5444±0.0868 |
| Warpformer | 0.0449±0.0010 | 0.1228±0.0018 | 0.1888±0.0598 | 0.2939±0.0591 | 0.3056±0.0011 | 0.3661±0.0016 | 0.4302±0.0035 | 0.4025±0.0014 |
| tPatchGNN | 0.0443±0.0009 | 0.1247±0.0031 | 0.1885±0.0403 | 0.3084±0.0479 | 0.3133±0.0053 | 0.3697±0.0049 | 0.4431±0.0115 | 0.4077±0.0088 |
| GraFITi | 0.0437±0.0005 | 0.1221±0.0017 | 0.1691±0.0093 | 0.2777±0.0248 | 0.3075±0.0015 | 0.3637±0.0036 | 0.4359±0.0455 | 0.4142±0.0297 |
| Hi-Patch | 0.0435±0.0002 | 0.1204±0.0009 | 0.1749±0.0268 | 0.2717±0.0216 | 0.3071±0.0029 | 0.3675±0.0042 | 0.4279±0.0010 | 0.4033±0.0032 |
| **ASTGI (Ours)** | **0.0412±0.0005** | **0.1181±0.0010** | **0.1608±0.0110** | **0.2597±0.0155** | **0.3004±0.0008** | **0.3589±0.0015** | **0.3909±0.0017** | **0.3852±0.0004** |

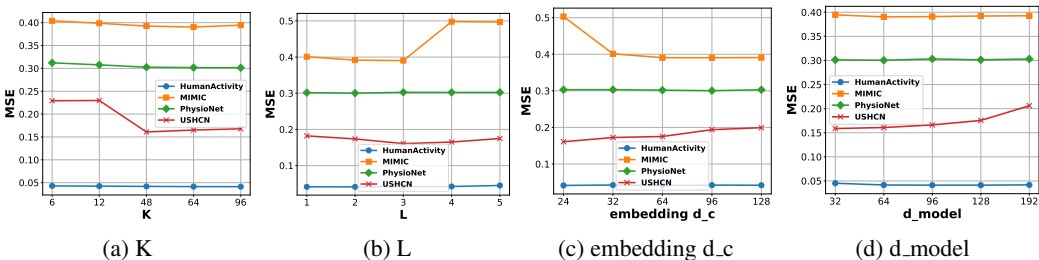

| (a) K | (b) L | (c) embedding d_c | (d) d_model |
|---|---|---|---|

Figure 4: Parameter sensitivity studies of main hyper-parameters in ASTGI.

PhysioNet), biomechanics (Human Activity), and climate science (USHCN), ASTGI consistently outperforms all competing methods, highlighting its strong generalization capability and robustness.

The superior performance of ASTGI can be attributed to its innovative modeling paradigm, which effectively addresses the two core challenges outlined in the introduction. First, by directly representing discrete observations as Spatio-Temporal Points, ASTGI completely avoids data alignment or interpolation, thus preserving the integrity of the original information. Second, and more critically, it replaces interaction structures that rely on fixed rules with a data-driven Neighborhood-Adaptive Graph Construction mechanism. This allows the model to adaptively identify the most relevant neighbors for each observation point and to capture complex dependencies across time and variables through the subsequent Spatio-Temporal Dynamic Propagation process.

### 4.3 PARAMETER SENSITIVITY

To investigate the ASTGI framework's dependency on key hyperparameters, we performed a sensitivity analysis on the number of candidate neighbors ($K$), number of propagation layers ($L$), model hidden dimension ($d_{model}$), and channel embedding dimension ($d_c$)—see Figure 4. The analysis reveals that the model's performance is not overly sensitive to these parameters and remains stable within a reasonable range. Specifically: (1) $K$: Performance stabilizes once $K$ reaches a threshold sufficient to capture key information. This indicates that our Neighborhood-Adaptive Graph Construction mechanism can effectively identify and utilize the most important neighbors without overly relying on a precise or large neighborhood size. (2) $L$: Generally, a small number of layers is sufficient to capture complex spatio-temporal dependencies. Too many layers can introduce the risk of over-smoothing on complex datasets, leading to a slight decrease in performance. (3) $d_{model}$ and $d_c$: The optimal choice is directly related to the intrinsic complexity of the dataset. The model

Table 3: Ablation study of ASTGI components. Results are reported in **MSE** (mean ± std). The performance of our full model is highlighted in **bold**.

| Model / Dataset | Human Activity | USHCN | PhysioNet | MIMIC |
|---|---|---|---|---|
| w/o Learned Coordinates | 0.0421±0.0005 | 0.1838±0.0202 | 0.3034±0.0020 | 0.4057±0.0019 |
| w/o Adaptive Graph | 0.0421±0.0002 | 0.1830±0.0123 | 0.3164±0.0009 | 0.4065±0.0054 |
| w/o Relation-Aware | 0.0418±0.0008 | 0.1930±0.0246 | 0.3072±0.0030 | 0.4194±0.0034 |
| rp. Mean Pooling | 0.0870±0.0056 | 0.1699±0.0105 | 0.4826±0.0150 | 0.8807±0.0036 |
| **ASTGI (ours)** | **0.0412±0.0005** | **0.1607±0.0110** | **0.3004±0.0008** | **0.3909±0.0017** |

Table 4: Ablation study of ASTGI components. Results are reported in **MAE** (mean ± std). The performance of our full model is highlighted in **bold**.

| Model / Dataset | Human Activity | USHCN | PhysioNet | MIMIC |
|---|---|---|---|---|
| w/o Learned Coordinates | 0.1193±0.0013 | 0.2769±0.0153 | 0.3620±0.0028 | 0.3932±0.0044 |
| w/o Adaptive Graph | 0.1194±0.0007 | 0.2892±0.0147 | 0.3712±0.0034 | 0.3948±0.0067 |
| w/o Relation-Aware | 0.1210±0.0038 | 0.3002±0.0221 | 0.3664±0.0029 | 0.4023±0.0023 |
| rp. Mean Pooling | 0.1973±0.0083 | 0.2684±0.0146 | 0.5028±0.0110 | 0.6492±0.0062 |
| **ASTGI (ours)** | **0.1181±0.0010** | **0.2597±0.0155** | **0.3589±0.0015** | **0.3852±0.0004** |

requires sufficient representational power to encode data patterns, but overly high dimensions can increase the risk of overfitting, especially for datasets with fewer variables.

### 4.4 ABLATION STUDIES

To verify the effectiveness of each component in the ASTGI framework, we conduct a series of ablation studies—see Table 3 for the MSE metric and Table 4 for the MAE metric. We draw the following four key conclusions. (1) the learnable coordinate space enhances representation power. Replacing the learnable time and channel embeddings with fixed, non-parametric encodings leads to a significant drop in performance. This indicates that an adaptively learned metric space is crucial for capturing the unique non-linear patterns and inter-variable correlations within the data. (2) the data-driven adaptive graph is superior to a static structure. Degrading the basis for neighborhood search from the learned spatio-temporal coordinates to the original timestamps results in a notable performance decline. This demonstrates that adaptively discovering neighbors in the learned metric space is more effective than relying on fixed rules of temporal proximity. (3) the relation-aware propagation mechanism improves interaction precision. Removing the spatio-temporal displacement vector ($\boldsymbol{p}_i - \boldsymbol{p}_j$) as input when calculating interaction weights and messages causes a significant performance degradation. This highlights that modulating information based on the relative spatio-temporal position of neighbors is vital for capturing relation-dependent dynamics. (4) the dedicated query aggregation mechanism outperforms simple pooling. Replacing the interaction-weighted fusion in the prediction stage with simple neighborhood average pooling leads to a substantial drop in performance. This confirms that differentially weighting neighbor information based on its spatio-temporal relationship to the query point allows the model to focus more effectively on critical information during prediction.

## 5 CONCLUSION

This paper introduces the ASTGI framework to address the core challenges of information distortion and static interactions in IMTSF. At its core, the framework maps each discrete observation directly into a learnable spatio-temporal embedding space. This design fundamentally avoids the information distortion caused by data preprocessing steps like interpolation or alignment. More critically, ASTGI abandons predefined interaction structures. Instead, it constructs a data-driven causal neighborhood graph for each point within the embedding space and employs a relation-aware propagation mechanism to precisely model complex dynamics that span across time and variables. Experimental results across multiple public datasets consistently show that ASTGI achieves significantly higher prediction accuracy than state-of-the-art methods, demonstrating that this adaptive graph interaction paradigm is an effective and promising new direction for solving irregular time series problems.

## ACKNOWLEDGEMENTS

This work was partially supported by the National Natural Science Foundation of China (No. 62372179, No. 62472174) and the Fundamental Research Funds for the Central Universities. Jilin Hu is the corresponding author of the work.

## ETHICS STATEMENT

Our work exclusively uses publicly available benchmark datasets that contain no personally identifiable information. No human subjects are involved in this research.

## REPRODUCIBILITY STATEMENT

The promise that all experimental results can be reproduced. We have released our model code in an anonymous repository: https://github.com/decisionintelligence/ASTGI.

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

## A   THE USE OF LARGE LANGUAGE MODELS (LLMs)

We do not use Large Language Models in our methodology and writing.

## B   DETAILED EXPERIMENTAL SETUP

### B.1   DATASETS DETAILS

In this section, we provide a detailed description of the four public datasets used in our experiments, including their sources, characteristics, and the specific preprocessing steps applied.

**MIMIC** is a large, freely-accessible critical care database (Johnson et al., 2016). It contains de-identified health data from patients who stayed in intensive care units (ICUs) at the Beth Israel Deaconess Medical Center between 2001 and 2012. The dataset is highly detailed, including vital signs, medications, and lab measurements. For our experiments, we use the clinical time series data from the first 48 hours of each patient's ICU stay. The MIMIC dataset contains 21,250 samples with 96 variables.

**PhysioNet** dataset is another valuable resource for clinical time series analysis (Johnson et al., 2014). It was released for a challenge to predict the in-hospital mortality of ICU patients. The dataset includes records from 12,000 ICU stays, with each record consisting of a multivariate time series of measurements from the first 48 hours. It comprises 11,981 samples and 36 variables, such as serum glucose and heart rate.

**Human Activity** dataset from the UCI Machine Learning Repository is used for research in human activity recognition (Rubanova et al., 2019). It contains data from sensors placed on the ankles, belt, and chest of five individuals performing various activities. The dataset includes 1,359 samples and 12 variables representing 3D positions. The data is naturally irregular because the sensors record information at slightly different time intervals.

**USHCN** (United States Historical Climatology Network) dataset offers long-term climate data from weather stations across the United States, covering over 150 years (Menne et al., 2015). It is a key resource for studying climate change. The dataset includes 1,114 samples and 5 variables, such as daily maximum and minimum temperatures and precipitation. Although data is recorded daily, missing observations are common, which makes it suitable for irregular time series analysis. In our study, we use a subset of the data from a 4-year period between 1996 and 2000, following the approach of previous work.

### B.2   BASELINE MODEL DETAILS

**IMTS CLASSIFICATION/IMPUTATION MODELS**

For all baseline models originally designed for classification, we replace the final Softmax layer with a linear layer to adapt them for the forecasting task.

**PRIMENET (CHOWDHURY ET AL., 2023)**

PrimeNet is a pre-training model for IMTS. Our experiments load its official pre-trained weights and fine-tune it on each dataset. The model's patch length varies by dataset, the number of heads is set to 1, and the learning rate is $1 \times 10^{-4}$.

**SEFT (HORN ET AL., 2020)**

SeFT processes all observation points in a time series as an unordered set. The model consists of 2 layers, a dropout rate of 0.1, and a learning rate of $1 \times 10^{-3}$.

**mTAN (SHUKLA & MARLIN, 2021)**

mTAN utilizes a multi-time attention mechanism to map features from an irregular series onto a fixed set of reference points. The number of reference points is set to 32 for the MIMIC dataset and defaults to 8 for others. The learning rate is $1 \times 10^{-3}$.

**GRU-D** (CHE ET AL., 2016)

GRU-D is an adaptation of the Gated Recurrent Unit (GRU) for IMTS with missing values. In our experiments, a learning rate of $1 \times 10^{-3}$ is used for this model.

**RAINDROP** (ZHANG ET AL., 2022)

Raindrop is a graph attention model for IMTS. Its hidden dimension is 32. The learning rate is $1 \times 10^{-3}$ for the HumanActivity dataset and $1 \times 10^{-4}$ for others. The number of heads is 4 for the HumanActivity and USHCN datasets and defaults to a different value for others.

**WARPFORMER** (ZHANG ET AL., 2023)

Warpformer uses a warping technique for multi-scale modeling. It has a hidden dimension of 256, 4 attention heads, a dropout rate of 0, and 2 layers. The learning rate is $1 \times 10^{-3}$.

**IMTS FORECASTING MODELS**

**NEURALFLOWS** (BILOS ET AL., 2021)

NeuralFlows is a model based on ordinary differential equations (ODEs). It includes 2 flow layers, a latent dimension of 20, a time encoding hidden dimension of 8, and uses 3 hidden layers. The learning rate is $1 \times 10^{-3}$.

**CRU** (SCHIRMER ET AL., 2022)

The CRU model uses continuous recurrent units to handle irregular time series. Its hidden dimension is set to 20, and the learning rate is $1 \times 10^{-3}$.

**GNEURALFLOW** (MERCATALI ET AL., 2024)

GNeuralFlow enhances NeuralFlows by incorporating graph neural networks. It uses a ResNet as its flow model with 2 flow layers. The input latent dimension is 20, the time encoding hidden dimension is 8, and it has 3 hidden layers. The learning rate is $1 \times 10^{-3}$.

**tPATCHGNN** (ZHANG ET AL., 2024B)

tPatchGNN first processes an IMTS into patches and then uses a graph neural network for forecasting. The patch length varies depending on the dataset. The number of heads is set to 1, and the learning rate is $1 \times 10^{-3}$.

**GRAFITI** (YALAVARTHI ET AL., 2024)

GraFITi uses bipartite graphs to represent irregular time series. Its latent dimension is 256 for the MIMIC dataset and 128 for others. The number of layers is set to 4 for MIMIC and USHCN, and 2 for the remaining datasets. The learning rate is $1 \times 10^{-3}$.

**HI-PATCH** (LUO ET AL., 2025)

Hi-Patch is a patch-based hierarchical Transformer model. Its hidden dimension, number of heads, and patch length are specifically set for each dataset. The learning rate is $5 \times 10^{-4}$ for the PhysioNet dataset and $1 \times 10^{-3}$ for others.

## C   QUALITATIVE ANALYSIS OF LEARNED INTERACTION GRAPHS

To validate the interpretability and effectiveness of the proposed *Neighborhood-Adaptive Graph Construction* module, we visualize the learned causal graph for a randomly selected sample from the MIMIC test set.

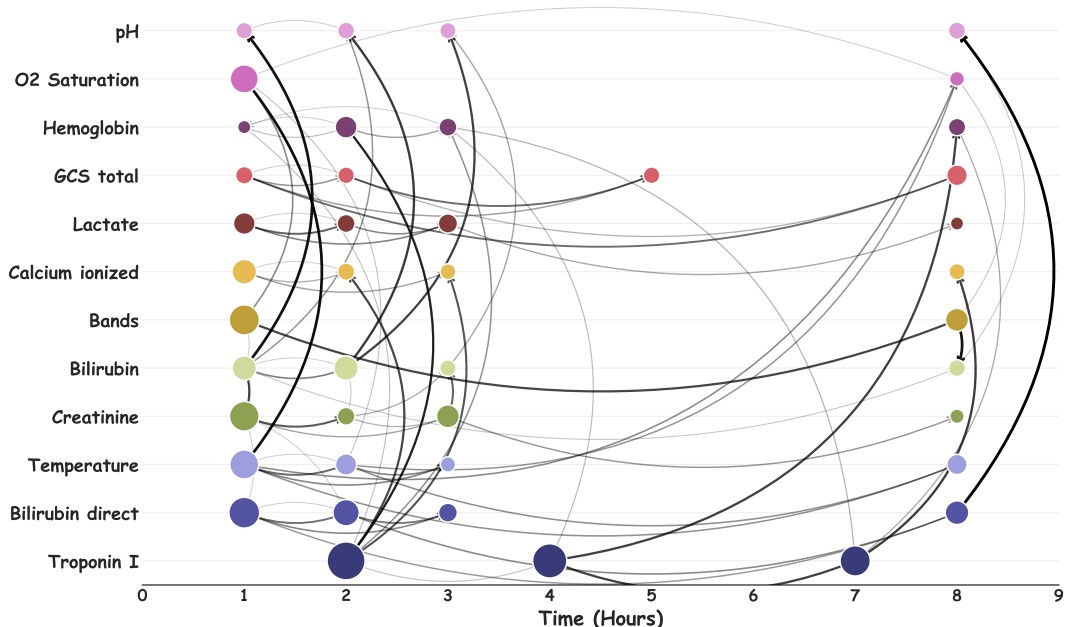

Figure 5: **Visualization of the adaptively learned causal graph.** The plot displays the interactions between observation points for a sample from the MIMIC dataset. The x-axis represents time (hours), and the y-axis represents different variables. Arrows indicate the direction of information flow (from history to query). The model successfully captures **(1)** synchronous correlations between variables (e.g., Bilirubin Direct and pH at $t = 8$), **(2)** long-range temporal dependencies (e.g., Troponin I self-connection $t = 4 \rightarrow t = 7$), and **(3)** cross-variable lagged effects. This confirms that ASTGI adaptively constructs a sparse and meaningful interaction topology.

Figure 5 illustrates the inference process. In this graph: (1) **Nodes** represent discrete observations, positioned horizontally by timestamp (Time) and vertically by variable type. (2) **Node Size** is proportional to the cumulative attention weight the node receives, indicating its importance in the current inference context. (3) **Edges** represent the learned attention scores. Darker and thicker lines indicate stronger dependencies. To ensure visual clarity, we only visualize the top-K strongest incoming edges for each node.

As shown in Figure 5, the learned structure exhibits three distinct patterns that align with the characteristics of irregular multivariate time series: (1) **Synchronous Multivariate Correlations:** The model frequently establishes strong connections between different variables observed at the same timestamp. For example, at $t = 8$, a significant interaction is observed between *Bilirubin Direct* and *pH* (indicated by the thick connecting line). This demonstrates that ASTGI can effectively leverage synchronous co-occurrence information to reconstruct the system state, bypassing the need for manual time alignment. (2) **Direct Long-Range Dependencies:** Unlike recurrent models that propagate information step-by-step, ASTGI enables direct information propagation across non-adjacent timestamps. A clear example is the variable *Troponin I*, where the observation at $t = 4$ strongly connects to the subsequent observation at $t = 7$. This mechanism allows the model to retrieve critical historical information directly, thereby mitigating the long-term dependency issues common in recurrent architectures. (3) **Cross-Variable Temporal Impact:** The graph also captures complex lagged dependencies across different variables. We observe that the state of *Troponin I* at $t = 4$ exerts a notable influence on *Hemoglobin* at $t = 8$. This suggests that the model has learned to associate early anomalies in specific physiological indicators with delayed responses in others, capturing the systemic dynamics of the underlying process.

# D    VISUALIZATION OF LEARNED SPATIO-TEMPORAL EMBEDDINGS

To intuitively understand how ASTGI represents discrete irregular observations, we provide a visualization analysis of the learned spatio-temporal coordinate space. We randomly selected a test sample from the MIMIC dataset and extracted the coordinate vectors $\boldsymbol{p}_i = \boldsymbol{e}_{c_i} \oplus \boldsymbol{e}_{t_i}$ for all its observation points. We then utilized t-SNE to project these high-dimensional coordinates into a 2D space.

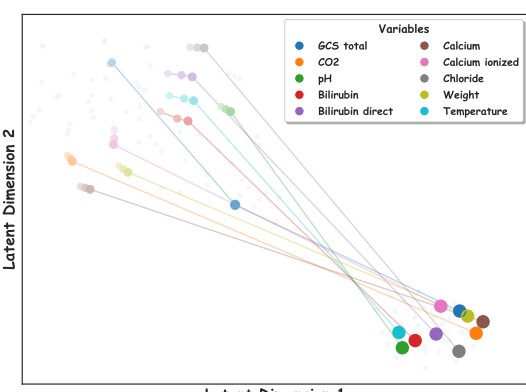

Figure 6 illustrates the resulting embedding structure. To clearly visualize the temporal evolution within the embedding space, we apply a time-dependent color gradient to the observation points of each variable. Specifically, lighter shades represent earlier observations, while darker and more saturated shades indicate later timestamps. The visualization highlights three key characteristics: (1) **Variable Distinctiveness:** Observations corresponding to different variables (represented by different colors) form distinct, well-separated clusters. This demonstrates that the learnable Channel Embedding effectively captures the unique semantic identities of different physiological indicators. (2) **Temporal Continuity:** For any given variable, the observation points do not collapse into a single spot but instead form a continuous trajectory. The smooth transition from light to dark colors confirms that our Time Encoding successfully preserves the sequential order and temporal intervals within the embedding space. (3) **Validity for Adaptive Graph Construction:** The combination of variable clustering and temporal trajectories creates a structured metric space. In this space, the Euclidean distance—used by our Neighborhood-Adaptive Graph Construction module—naturally prioritizes neighbors that are semantically correlated and temporally relevant. This validates our design choice of replacing fixed interaction graphs with dynamic k-NN search in this learned space.

Figure 6: **Visualization of the learned spatio-temporal embedding space.** We visualize the coordinates $\boldsymbol{p}_i$ of observations from a MIMIC sample using t-SNE. Points are colored by variable types. The temporal evolution is indicated by the color intensity: for each variable, the color transitions from light (early time) to dark (late time). The clear clustering and continuous gradients demonstrate that ASTGI effectively encodes both variable semantics and dynamic temporal patterns.

