# OpenReview forum: "ASTGI: Adaptive Spatio-Temporal Graph Interactions for Irregular Multivariate Time Series Forecasting"
_ICLR.cc/2026/Conference — ICLR 2026 Poster_

### Official Review · Reviewer_FrHL · 2025-10-28

**Soundness:** 3
**Presentation:** 4
**Contribution:** 3
**Rating:** 6
**Confidence:** 4

**Summary:**

This paper addresses two fundamental challenges in Irregular Multivariate Time Series Forecasting (IMTSF): 1) information distortion caused by data preprocessing like interpolation, and 2) the inability of static interaction structures to capture complex, dynamic dependencies. To tackle these issues, the paper proposes a novel framework named ASTGI. The framework preserves raw information by directly embedding each discrete observation into a learnable spatio-temporal space. It then adaptively constructs a causal graph for each point to capture dynamic relationships. Finally, it performs prediction through relation-aware message passing on these graphs. Experimental results on four public datasets show that ASTGI significantly outperforms a wide range of state-of-the-art baselines.

**Strengths:**

1.	It solves the information distortion problem in irregular time series modeling by directly mapping discrete observations to a learnable continuous space, avoiding any preprocessing.
2.	Its data-driven, adaptive graph construction mechanism discards all prior assumptions about interactions, enabling the model to capture highly complex and dynamic dependencies across time and variables.
3.	It seamlessly unifies the prediction task into its neighborhood aggregation framework, showing a high degree of conceptual consistency and design elegance that many other methods lack.
4.	It consistently outperforms a large number of strong baselines across multiple diverse benchmarks, providing powerful evidence of the framework's superior performance and broad applicability.

**Weaknesses:**

1.	An analysis of the model's computational complexity and its scalability on larger-scale datasets is not provided, making it difficult to assess its deployment potential in resource-constrained environments.
2.	The motivation for using a unified hyperparameter K across different modules is not sufficiently clarified; setting them separately might offer further optimization for neighborhood aggregation at different stages.
3.	After multi-layer graph propagation, the initial observation $x_i$'s magnitude information could be diluted, affecting final regression accuracy. The paper lacks discussion on mitigating this risk.

**Questions:**

1.	Regarding the model's computational efficiency, what is the theoretical computational complexity of constructing the adaptive graph for each observation point, and are there practical strategies to optimize this process?
2.	Regarding the hyperparameter K, what was the design consideration for using the same value for both the information propagation and the final prediction stages?
3.	If the K-nearest neighbors in the learned space are all in the future, the causal mask results in zero neighbors. How does the model handle this "neighbor starvation"?
4.	Why use a separate scoring network for the final prediction, instead of reusing the one from the propagation phase? What's the advantage of this design?

**Details Of Ethics Concerns:**

None.

---

> ### Author Response · Authors · 2025-11-21
> **Response to Reviewer FrHL (Part I)**
>
> **Response to W1: Computational complexity analysis and scalability optimization.**
>
> We acknowledge the importance of computational efficiency. Our ASTGI framework achieves high scalability and deployment potential through two key mechanisms inherent in its design and implementation:
> 1.  **Efficiency via Sparsity:** Unlike grid-based methods (e.g., Transformers) that compute interactions over the entire $L \times V$ grid regardless of missing values, ASTGI models discrete observation points directly. Since irregular time series are typically highly sparse (where the number of observed points $N_{obs} \ll L \times V$), our computational cost depends solely on the actual observations, avoiding redundant computations on missing data.
> 2.  **Accelerated Neighbor Search:** While the theoretical worst-case complexity for graph construction is $O(N^2)$, our implementation integrates the **FAISS** library to perform approximate nearest neighbor search. This optimization reduces the complexity from $O(N^2)$ to **$O(N \log N)$**, ensuring the model scales efficiently even on large-scale datasets with extensive historical observations.
>
> **Response to W2: Justification for the unified hyperparameter K.**
>
> We employ a unified $K$ across the propagation and prediction modules to ensure **topological consistency**. The nearest neighbors in the embedding space represent intrinsic causal dependencies governed by the underlying system dynamics, which should remain invariant regardless of the processing stage (feature update vs. final regression). Using distinct $K$ values would create a discrepancy between the context used for learning representations and the context used for inference, potentially degrading performance. Furthermore, our parameter sensitivity analysis (**Fig. 4a**) demonstrates that performance stabilizes once $K$ reaches a threshold sufficient to cover the local context (e.g., $K=32$). Therefore, a unified $K$ provides a robust and parsimonious solution without introducing unnecessary search space or overfitting risks.
>
> **Response to W3: Mitigation of information dilution via residual connections.**
>
> The ASTGI framework explicitly prevents the dilution of original magnitude information through the **residual connection design** in the Update Function (refer to **Eq. 6**). The update formula, $\boldsymbol{h}_i^{(l+1)} = \text{LayerNorm}(\boldsymbol{h}_i^{(l)} + \text{MLP}(\boldsymbol{m}_i^{(l)}))$, enforces the model to learn residual updates on top of the identity mapping. This ensures that the original state $\boldsymbol{h}_i^{(l)}$ is preserved and mathematically prevents the information degradation of initial observation values during propagation. Additionally, our ablation study (**Fig. 4b**) indicates that the model achieves optimal performance with a shallow architecture ($L=2$ to $4$), which naturally avoids the over-smoothing problem often associated with deep GNNs and preserves the distinctiveness of the raw observations.

---

> ### Author Response · Authors · 2025-11-21
> **Response to Reviewer FrHL (Part II)**
>
> **Response to Q1: Theoretical complexity and practical optimization strategies.**
>
> The theoretical baseline complexity for calculating pairwise distances among $N$ observation points is $O(N^2 \cdot (d_t + d_c))$. However, our ASTGI framework implements two critical strategies to optimize this in practice:
> 1.  **Efficiency via Sparsity:** Unlike grid-based models that process full $L \times V$ matrices, our point-based approach only computes interactions for valid observations ($N_{obs}$). Since irregular time series are inherently sparse ($N_{obs} \ll L \times V$), this naturally minimizes the computational load.
> 2.  **Algorithmic Acceleration:** As implemented in our code, we utilize **Approximate Nearest Neighbor (ANN)** search via the FAISS library. This reduces the search complexity from quadratic $O(N^2)$ to log-linear **$O(N \log N)$**. Furthermore, the distance computations are highly vectorized and GPU-parallelized, ensuring high throughput even for large-scale datasets.
>
> **Response to Q2: Design rationale for the unified hyperparameter K.**
>
> The decision to use a unified $K$ is driven by **topological consistency**. We premise that the learned embedding space defines the intrinsic causal structure of the system. Using the same $K$ ensures that the **receptive field** remains consistent across both the feature learning (propagation) and inference (prediction) phases. If distinct $K$ values were used (e.g., a large $K$ for propagation and a small $K$ for prediction), the model would suffer from a context mismatch, where information aggregated during feature updates is discarded during the final regression, leading to suboptimal predictions.
>
> **Response to Q3: Handling of empty neighborhoods via self-loops and residuals.**
>
> The model effectively handles scenarios with limited historical context through two mechanisms:
> 1.  **Inherent Self-Loops:** The causal mask is defined as $t_j \le t_i$. This inequality ensures that every observation point is a valid neighbor to itself. Therefore, even the first point in a sequence retrieves itself as a neighbor, preventing an empty set.
> 2.  **Residual Protection:** Even in the hypothetical absence of informative neighbors, the **residual connection** in the update function (Eq. 6), defined as $\boldsymbol{h}^{(l+1)} = \text{LayerNorm}(\boldsymbol{h}^{(l)} + \text{MLP}(\boldsymbol{m}^{(l)}))$, acts as a safeguard. It ensures that the point's current state $\boldsymbol{h}^{(l)}$ is preserved and propagated to the next layer, preventing information loss or numerical instability.
>
> **Response to Q4: Advantages of separate scoring networks.**
>
> We employ separate scoring networks for propagation and prediction to achieve **task decoupling**. The propagation module focuses on **Latent-to-Latent** dynamics, learning how states evolve over time to capture complex temporal dependencies. In contrast, the prediction module performs a **Latent-to-Value** mapping, focusing on regressing the specific target value from the aggregated features. Separating these parameters allows the model to learn distinct attention distributions optimized for these different objectives—such as focusing on long-term trends for state updates versus immediate local contexts for value regression—thereby avoiding gradient interference and improving overall accuracy.

---

> > ### Comment · Reviewer_FrHL · 2025-11-22
> >
> > Thank the authors for their detailed response. The clarifications regarding the three weaknesses and the four questions addressed most of the concerns, especially the explanation of the model’s computational complexity. Based on these improvements, I have decided to raise my score from 6 to 8. Wishing the authors good luck.

---

> > > ### Author Response · Authors · 2025-11-22
> > > **Thank you so much for your encouraging feedback and for your support toward the acceptance of our paper.**
> > >
> > > Dear Reviewer FrHL:
> > >
> > > Thank you so much for your encouraging feedback and for your support toward the acceptance of our paper. We sincerely appreciate your time and constructive comments throughout the review process.
> > >
> > > Best regards,
> > >
> > > Authors

---

### Official Review · Reviewer_LeMp · 2025-10-30

**Soundness:** 2
**Presentation:** 3
**Contribution:** 2
**Rating:** 4
**Confidence:** 4

**Summary:**

The paper proposes ASTGI, a novel framework for Irregular Multivariate Time Series Forecasting (IMTSF), which addresses two core challenges: (1) avoiding information distortion caused by interpolation or alignment, and (2) capturing complex dynamic dependencies between observations. ASTGI introduces a learnable spatio-temporal embedding space where each observation is represented as a point. It then adaptively constructs causal interaction graphs via nearest-neighbor search and performs relation-aware message propagation to model dependencies. A query-based prediction module is used to forecast future values. Extensive experiments on four public datasets show that ASTGI outperforms state-of-the-art baselines, and ablation studies confirm the importance of each component.

**Strengths:**

1. IMTSF is an interesting topic. The idea of representing IMTS as points in a learnable spatio-temporal space and adaptively constructing causal graphs effectively bridges the gap between raw data representation and dynamic dependency modeling.
2. The methodology is clearly described, and the authors have provided code and detailed experimental setups, which supports reproducibility.
3. The paper includes experiments on four diverse datasets (healthcare, climate, activity) and compares against 12 strong baselines. The results are convincing and statistically sound.

**Weaknesses:**

1. The connections between the two key challenges are not clearly explained.
2. In the Candidate Neighborhood Identification module, the choice of Euclidean distance in the spatiotemporal coordinate space lacks sufficient justification.
3. The technical contributions of the Neighborhood-adaptive Graph Construction and Spatiotemporal Dynamic Propagation modules are limited.
4. The Related Work section lacks an overarching overview.
5. There is a typo in Equation (8): $a_{q,i}$ should be $s_{q,i}$.

**Questions:**

1. What is the relationship between raw information representation and dynamic dependency modeling in IMTS?
2. Regarding Candidate Neighborhood Identification:
(1) Why is distance measured in the coordinate space rather than the feature space?
(2) Why is Euclidean distance used instead of Cosine similarity?
(3) Why is normalization not applied? Does the embedding norm influence performance?
(4) Why are points with distant coordinates discarded? Could this remove long-range dependencies?
(5) How is the top-k operation differentiated? Might it suppress useful neighbors during early training stages?
3. What is the novelty of the Spatiotemporal Dynamic Propagation module compared to existing message-passing mechanisms in Graph Neural Networks?
4. Both the raw interaction score in Equation (3) and the message in Equation (4) incorporate the relative positional embedding $p_i - p_j$. How do their roles differ?
5. In Line 322, why does such a separation improve flexibility and accuracy?
6. Can the authors provide visualized analyses of the spatiotemporal points in the embedding space?

---

> ### Author Response · Authors · 2025-11-21
> **Response to Reviewer LeMp (Part I)**
>
> **Response to W1: The connections between the two key challenges.**
>
> We thank the reviewer for this insightful comment. We agree that clarifying the relationship between these two challenges is crucial. There is an inherent **progressive and causal relationship** between them: **Accurate representation (Challenge 1) is the prerequisite and foundation for effective dependency modeling (Challenge 2).**
>
> *   **Foundational Role of Challenge 1:** The raw sampling pattern of irregular time series contains vital information about the system's underlying dynamics. If the representation step introduces distortion (e.g., through interpolation or misalignment), it corrupts this intrinsic information.
> *   **Dependency of Challenge 2:** Consequently, any attempt to capture dynamic dependencies (Challenge 2) based on a distorted representation would be modeling noise or artifacts rather than the true system dynamics, limiting predictive performance.
>
> Our ASTGI framework is explicitly designed to follow this logic: we *first* solve Challenge 1 by mapping discrete observations to a continuous embedding space to preserve raw information, and *then* solve Challenge 2 by performing adaptive graph convolution within this accurate metric space.
>
> We have revised the **Introduction** to explicitly articulate this connection. Please refer to the revised paper where the added text is marked in blue.
>
> **Response to W2: Justification for Euclidean distance in spatiotemporal coordinate space.**
>
> We employ Euclidean distance based on the specific geometric properties and design goals of the Spatio-Temporal Coordinate Space:
>
> 1.  **Consistency with Metric Space Design:** The coordinate space is explicitly constructed as a **metric space** to encode spatio-temporal proximity, where embeddings serve as positions rather than semantic features. In this geometric context, Euclidean distance is the canonical metric for defining neighborhoods (i.e., $k$-NN). It ensures that the graph construction adheres to the physical intuition of locality within the learned latent structure, which is different from calculating semantic similarity in high-level feature spaces (where dot-product is typically used).
> 2.  **Preservation of Magnitude Information:** Euclidean distance accounts for both the direction and magnitude of the embedding vectors. In our framework, the magnitude of the learnable channel and time embeddings encodes meaningful variations in the data structure. Using metrics like Cosine similarity, which normalize vectors, would discard this intrinsic magnitude information, thereby reducing the representational capacity of the coordinate space.
>
> **Response to W3: Technical contributions of the graph construction and propagation modules.**
>
> We respectfully clarify that the contribution of these modules lies in their specific adaptation to solve the limitations of current IMTS forecasting methods, rather than being generic applications of GNNs.
>
> 1.  **Departure from Static Heuristics (Graph Construction):** Prior GNN-based IMTS methods (e.g., tPatchGNN, HyperIMTS) universally rely on **predefined, static heuristic rules** (such as temporal adjacency or variable identity) to define graph topology. This imposes a fixed inductive bias that restricts the model's ability to capture evolving dependencies. In contrast, ASTGI is the first framework in this domain to implement a **fully adaptive, data-driven topology**. By dynamically generating causal neighborhoods based on proximity in the learned metric space, our approach removes the reliance on manual heuristics, allowing the model to autonomously discover complex, non-trivial dependencies that static rules fail to capture.
> 2.  **Geometry-Aware Message Passing (Dynamic Propagation):** The propagation module is not a standard GNN layer but is specifically engineered to exploit the geometric properties of the constructed graph. Unlike generic GNNs that aggregate features based on binary connectivity, our module explicitly incorporates the **relative spatio-temporal position vector** $(p_i - p_j)$ into the message generation and weighting mechanism. This design creates a **geometry-aware propagation** scheme where information flow is modulated by the precise metric relationships defined in the coordinate space, ensuring that the feature evolution is strictly consistent with the learned spatio-temporal structure.

---

> ### Author Response · Authors · 2025-11-21
> **Response to Reviewer LeMp (Part II)**
>
> **Response to W4: Lack of an overarching overview in Related Work.**
>
> We appreciate this constructive suggestion. We agree that a high-level summary is essential for clarifying the research landscape. We have revised **Section 2 (Related Work)** to include an introductory paragraph at the beginning. This overview systematically categorizes existing IMTS forecasting methods into two primary paradigms—**Structured Representation-based** and **Raw-Data-based**—outlining their core philosophies and limitations, thereby explicitly positioning our ASTGI framework within the broader literature. Please refer to the revised paper for the added text marked in blue.
>
> **Response to W5: Alleged typo in Equation (8).**
>
> We respectfully clarify that **Equation (8) is correct as written**. The symbol $a_{qi}$ denotes the **normalized interaction weights**, which are derived by applying the Softmax function to the raw association scores $s_{qi}$ (calculated in Eq. 7), as explicitly described in the text (Line 339). In the weighted fusion process (Eq. 8), it is mathematically necessary to use these normalized weights $a_{qi}$ (which sum to 1) rather than the raw, unbounded scores $s_{qi}$ to ensure a properly scaled and stable aggregation of neighbor information.
>
> **Response to Q1: The relationship between raw information representation and dynamic dependency modeling.**
>
> There exists an inherent progressive and causal relationship between these two aspects: **Accurate representation is the prerequisite and foundation for effective dependency modeling.**
>
> 1.  **Foundational Role of Representation:** The raw sampling pattern of irregular time series contains vital information about the system's underlying dynamics. If the representation step introduces distortion, such as through interpolation or misalignment, it corrupts this intrinsic information.
> 2.  **Dependency of Modeling:** Consequently, any subsequent attempt to capture dynamic dependencies based on a distorted representation would be modeling artifacts rather than the true system dynamics, limiting predictive performance.
>
> Our ASTGI framework strictly follows this logic: we first employ the Spatio-Temporal Point Representation to encode discrete observations without distortion, which then serves as the precise metric space for the subsequent Neighborhood-Adaptive Graph Construction and Dynamic Propagation.

---

> ### Author Response · Authors · 2025-11-21
> **Response to Reviewer LeMp (Part III)**
>
> **Response to Q2: Clarifications on Candidate Neighborhood Identification.**
>
> **(1) Coordinate Space vs. Feature Space:**
> We utilize the coordinate space to define a **stable metric structure** for graph construction. The coordinates represent the inherent spatio-temporal context of observations. In contrast, the feature vectors $h$ serve as the dynamic information state that evolves iteratively across layers. Defining neighbors in the feature space would cause the graph topology to fluctuate at every propagation step, leading to training instability and inconsistent dependency modeling.
>
> **(2) Euclidean Distance vs. Cosine Similarity:**
> Our objective is to construct a metric space where **geometric proximity** is defined by both direction and magnitude. Euclidean distance naturally captures this full geometric relationship. Conversely, Cosine similarity normalizes vectors, effectively discarding the **magnitude information**. In our framework, the magnitude of the learnable embeddings encodes critical discriminative information (e.g., variable importance or signal intensity), which would be lost if normalized.
>
> **(3) Normalization and Embedding Norm:**
> We intentionally avoid normalization to preserve the representational capacity of the embedding space. Normalization would constrain the embeddings to a hypersphere, limiting the model's ability to express complex structural relationships. The embedding norm directly influences performance by determining the neighborhood set during the $k$-NN search and serving as a distinct input feature within the relation-aware scoring mechanism.
>
> **(4) Long-range Dependencies:**
> We do not discard dependencies based on temporal distance. Crucially, the distance in our framework refers to the learned **metric distance**, not the absolute temporal interval. The model effectively learns to map temporally distant but semantically correlated points to proximal locations in the coordinate space, allowing the $k$-NN search to directly capture long-range dependencies. Furthermore, the stacking of multiple propagation layers expands the receptive field, enabling information to flow between distant nodes via multi-hop connections.
>
> **(5) Top-K Differentiation:**
> While the discrete Top-$k$ selection is non-differentiable, the coordinates $p$ and features $h$ of the selected neighbors are directly involved in the subsequent differentiable operations, specifically the **Relation-Aware Scoring (Eq. 2)** and **Message Function (Eq. 4)**. Consequently, gradients flow through these components to update the embeddings. During early training, the task loss drives the coordinate embeddings to cluster relevant points together, effectively creating a self-correcting mechanism that optimizes the neighborhood structure end-to-end.
>
> **Response to Q3: Novelty of the Spatiotemporal Dynamic Propagation module.**
>
> The novelty lies in our **Continuous Spatio-Temporal Displacement-Conditioned Message Generation**, which distinguishes ASTGI from standard message-passing mechanisms (e.g., GAT, GraphSAGE) in two critical aspects:
>
> 1.  **Content Modulation vs. Weight Adaptation:** Existing methods like GAT compute attention scores to weight the aggregation, yet the message content transmitted from a neighbor $j$ remains a fixed transformation of its feature $h_j$ (i.e., $W h_j$), independent of the receiver's spatial context. In contrast, ASTGI explicitly injects the relative position vector $(p_i - p_j)$ into the message generation function. This design enables the model to non-linearly modulate the feature content itself based on the geometric relationship, allowing for the precise modeling of distance-dependent dynamics such as continuous signal decay or evolution.
> 2.  **Continuous Metric Dependency:** Unlike standard temporal GNNs that rely on discrete time steps or scalar time intervals, our propagation operates directly within a continuous metric space. By mathematically conditioning the message flow on the vector difference in the learned coordinate space, the mechanism naturally accommodates irregular intervals and asynchronous dependencies without requiring time discretization, thereby preserving the integrity of the underlying continuous dynamics.

---

> ### Author Response · Authors · 2025-11-21
> **Response to Reviewer LeMp (Part IV)**
>
> **Response to Q4: Distinct roles of relative positional embedding in Eq. (3) and Eq. (4).**
>
> Although both equations utilize the relative position vector $p_i - p_j$, it serves distinct and complementary functional roles, enabled by the decoupled parameterization of $MLP_{score}$ and $MLP_{msg}$:
>
> 1.  **Structural Relevance Quantification in Eq. (3):** Here, the relative position acts as a **metric for relevance**. The $MLP_{score}$ maps the spatio-temporal displacement to a normalized scalar weight $a_{ij}$. Its primary function is to evaluate the compatibility or importance of the neighbor $j$ relative to the target $i$ within the metric space. This process effectively defines the soft topology of the graph by determining the intensity of the interaction.
> 2.  **Context-Dependent Feature Modulation in Eq. (4):** Here, the relative position serves as a **condition for state transformation**. The $MLP_{msg}$ uses the displacement vector to non-linearly modulate the neighbor's feature $h_j$. This allows the model to encode specific dynamical effects associated with the spatio-temporal shift (such as signal decay over time or phase shifts across variables) directly into the message vector. This process determines the specific semantic content of the information being propagated.
>
> **Response to Q5: Justification for parameter separation.**
>
> The parameter separation improves performance by decoupling the optimization of two distinct functional objectives:
>
> 1.  **Latent Representation Learning (Propagation):** The propagation module aims to evolve node features within the latent space to capture deep spatio-temporal dependencies. Its parameters must be optimized to sustain iterative information flow and maintain stable feature distributions across multiple layers without degradation.
> 2.  **Target Value Regression (Prediction):** The prediction module aims to project these high-dimensional latent representations onto the specific scalar target space. Its parameters must be optimized to minimize the regression error based on the aggregated context.
>
> Sharing parameters would force the same network weights to simultaneously satisfy both the recursive stability required for deep propagation and the discriminative mapping required for regression. This creates a competing optimization landscape. By separating them, we allow the propagation layers to focus exclusively on constructing rich representations, while the prediction head focuses exclusively on precise value readout, thereby enhancing both flexibility and accuracy.
>
>
> **Response to Q6: Visualized analyses of the spatiotemporal points.**
>
> We genuinely appreciate this insightful suggestion. Visualizing the latent space is indeed the best way to verify the effectiveness of our representation learning.
>
> In the revised paper, we have added a new **Appendix D** and **Figure 6** to provide a detailed visualization analysis. Specifically:
> 1.  We extracted the learned spatio-temporal coordinates $p_i = e_{c_i} \oplus e_{t_i}$ from a randomly selected sample in the MIMIC dataset.
> 2.  We utilized **t-SNE** to project these high-dimensional coordinates into a 2D space.
> 3.  To clearly illustrate the dynamic evolution, we employed a **time-dependent color gradient**: lighter shades represent earlier timestamps, while darker, saturated shades indicate later observations.
>
> The visualization reveals three critical properties that validate our design:
> *   **Semantic Disentanglement:** Observations from different variables form distinct, well-separated clusters, proving that the Channel Embedding successfully captures variable-specific semantics.
> *   **Temporal Continuity:** Within each variable cluster, points form a continuous trajectory (from light to dark) rather than a disordered cloud. This confirms that our Time Encoding effectively preserves the sequential order and temporal intervals.
> *   **Metric Space Validity:** The resulting structured space ensures that the Euclidean distance—used by our **Neighborhood-Adaptive Graph Construction**—naturally prioritizes neighbors that are both semantically correlated and temporally relevant.
>
> We believe this visualization provides strong empirical evidence supporting the rationale behind our adaptive graph mechanism.

---

### Official Review · Reviewer_2C1m · 2025-10-31

**Soundness:** 2
**Presentation:** 3
**Contribution:** 2
**Rating:** 2
**Confidence:** 3

**Summary:**

This paper proposes ASTGI, a framework that models irregular multivariate time series by representing each observation as a spatio-temporal point and dynamically constructing adaptive neighborhood graphs for message passing and forecasting.
The core idea is to treat each timepoint as a node and learn relations based on temporal and channel embeddings, followed by adaptive graph propagation and query-based prediction. Experimental results on several IMTS benchmarks demonstrate improvements over prior methods.

**Strengths:**

- The overall framework is clearly described and modularized, including point representation, adaptive neighbor discovery, and dynamic propagation.
- The authors conduct thorough ablation studies to justify several architectural design choices.

**Weaknesses:**

- A central concern is that the conceptual contribution is significantly overlapped with HyperIMTS (Li et al., ICML 2025). HyperIMTS also models each observation timepoint as a node and constructs a relational structure on top of it using temporal and channel information, enabling information aggregation over learned irregular-TS graph topology. Although the exact mechanisms differ (hypergraph vs nearest-neighbor graph), the core idea (treating each observation as a graph node and learning adaptive relational structure in irregular time series) is shared. The authors cite HyperIMTS in Introduction, but do not compare its performance in Experiment, which is problematic for a fair positioning of contributions.
- The paper motivates adaptive graph construction as a key novelty and advantage. However, the experimental section only evaluates forecasting performance and ablations, without providing any qualitative or quantitative analysis of the learned graph structures.

**Questions:**

Could the authors report training and inference wall-clock time and GPU memory comparison between ASTGI and key baselines (e.g., CRU, Warpformer, tPatchGNN, HyperIMTS if added)? Nearest-neighbor search among N nodes generally incurs at least $O(NK)$, and naive top-K selection may approach $O(N^2)$ if not optimized. For typical IMTS settings with thousands of points per sample, this can become computationally heavy.

---

> ### Author Response · Authors · 2025-11-21
> **Response to Reviewer 2C1m (Part I)**
>
> **Response to W1: Comparison with HyperIMTS and Conceptual Distinction**
>
> We thank the reviewer for highlighting this relevant work. We acknowledge that both ASTGI and HyperIMTS (Li et al., ICML 2025) model observations as nodes. However, we respectfully posit that there is **no significant contribution overlap**. The distinction is not merely a choice of mechanism (Hypergraph vs. NN Graph), but a fundamental shift in **how dependencies are discovered**, specifically in irregular regimes.
>
> **1. Conceptual Distinction: Static Heuristics vs. Adaptive Metric Learning**
>
> *   **Topology Construction (Fixed vs. Learned):** HyperIMTS relies on **fixed, pre-defined rules**, connecting nodes strictly based on shared timestamps or variable indices. This creates a rigid topology where cross-variable temporal dependencies (e.g., Var $A$ at $t_1$ $\to$ Var $B$ at $t_2$) must rely on multi-hop propagation. In contrast, ASTGI **learns the topology from scratch** in a continuous latent metric space. By performing dynamic nearest neighbor search, ASTGI establishes **direct, data-driven causal connections** between any two observations based on semantic proximity, bypassing the rigid structural constraints of fixed hyperedges.
>
> *   **Handling Irregularity (Alignment-Dependent vs. Alignment-Free):** This is the most critical difference.
>     *   **HyperIMTS:** As defined in Eq. 10-11 of their paper, HyperIMTS relies on "shared timestamps" ($T_{shared}$) to calculate dynamic attention. Crucially, when observations are unaligned (i.e., $T_{shared} \to 0$, which is common in sparse IMTS), their mechanism explicitly degenerates to a **static** variable-level correlation ($S_{var}$), effectively losing the ability to capture instance-specific dynamics.
>     *   **ASTGI:** We utilize a continuous spatio-temporal metric space ($p_i = e_{c_i} \oplus e_{t_i}$). By calculating distances in this space, ASTGI allows any observation to directly attend to relevant historical points **regardless of timestamp alignment**. This makes ASTGI inherently robust to high sparsity where HyperIMTS structurally fails.
>
> **2. Empirical Comparison**
>
> We have compared ASTGI with HyperIMTS using their official implementation under our unified experimental setting. As shown in the table below, ASTGI achieves superior or comparable performance to HyperIMTS across all datasets.
>
> Notably, on the **MIMIC** dataset, which is characterized by high sparsity and irregularity, ASTGI achieves a **9.1% reduction in MSE** compared to HyperIMTS. This empirically validates that our metric-space approach is superior to the alignment-based hypergraph in handling real-world irregularity.
>
> | Dataset | Human Activity (MSE) | Human Activity (MAE) | USHCN (MSE) | USHCN (MAE) | PhysioNet (MSE) | PhysioNet (MAE) | MIMIC (MSE) | MIMIC (MAE) |
> | :--- | :--- | :--- | :--- | :--- | :--- | :--- | :--- | :--- |
> | Warpformer | 0.0449±0.0010 | 0.1228±0.0018 | 0.1888±0.0598 | 0.2939±0.0591 | 0.3056±0.0011 | 0.3661±0.0016 | 0.4302±0.0035 | 0.4025±0.0014 |
> | tPatchGNN | 0.0443±0.0009 | 0.1247±0.0031 | 0.1885±0.0403 | 0.3084±0.0479 | 0.3133±0.0053 | 0.3697±0.0049 | 0.4431±0.0115 | 0.4077±0.0088 |
> | GraFITi | 0.0437±0.0005 | 0.1221±0.0017 | 0.1691±0.0093 | 0.2777±0.0248 | 0.3075±0.0015 | 0.3637±0.0036 | 0.4359±0.0455 | 0.4142±0.0297 |
> | Hi-Patch | 0.0435±0.0002 | 0.1204±0.0009 | 0.1749±0.0268 | 0.2717±0.0216 | 0.3071±0.0029 | 0.3675±0.0042 | 0.4279±0.0010 | 0.4033±0.0032 |
> | HyperIMTS | 0.0429±0.0007 | 0.1215±0.0013 | 0.1703±0.0091 | 0.2683±0.0125 | **0.3001±0.0011** | 0.3601±0.0019 | 0.4297±0.0027 | 0.3879±0.0009 |
> | **ASTGI (Ours)** | **0.0412±0.0005** | **0.1181±0.0010** | **0.1608±0.0110** | **0.2597±0.0155** | 0.3004±0.0008 | **0.3589±0.0015** | **0.3909±0.0017** | **0.3852±0.0004** |
>
> We believe these clarifications and results effectively distinguish our work from HyperIMTS and demonstrate the superiority of our adaptive graph interaction framework.

---

> ### Author Response · Authors · 2025-11-21
> **Response to Reviewer 2C1m (Part II)**
>
> **Response to W2: Qualitative analysis of the learned graph structures.**
>
> We thank the reviewer for this constructive suggestion. We agree that analyzing the learned graph is essential to verify whether ASTGI truly captures meaningful dependencies.
>
> In the revised paper, we have added **Appendix C** and **Figure 5** to qualitatively analyze the learned graph structures. We visualized the attention weights from the *Neighborhood-Adaptive Graph Construction* module for a random test sample from MIMIC. In the visualization, nodes represent discrete observations $(t, \text{variable})$, node sizes indicate aggregated importance, and edges represent the top-K learned interactions.
>
> The visualization provides three compelling insights into the model's behavior:
> 1.  **Synchronous Multivariate Correlations:** The model assigns high attention weights between different variables observed at the same (or very close) timestamps. For instance, we observe a strong connection between *Bilirubin Direct* and *pH* at $t=8$, indicating the model utilizes immediate co-occurrences to enrich context without manual alignment.
> 2.  **Direct Long-Range Dependencies:** Unlike RNNs that degrade over time, ASTGI establishes direct direct information propagation connections across long intervals. For example, *Troponin I* at $t=4$ directly attends to its future state at $t=7$, effectively preserving critical historical memory.
> 3.  **Cross-Variable Temporal Impact:** The graph reveals complex, lagged causalities. An anomaly in *Troponin I* at $t=4$ is shown to influence *Hemoglobin* levels at $t=8$, demonstrating the capability to model delayed systemic effects across different physiological indicators.
>
> These patterns confirm that ASTGI learns a sparse, semantically meaningful causal graph rather than a random or static structure.
>
> **Response to Q1: Computational Efficiency and Complexity Analysis**
>
> We appreciate the reviewer's scrutiny regarding computational complexity, which is indeed critical for scalability. To address this, we conducted a comprehensive benchmark on the **MIMIC** dataset comparing ASTGI with key baselines including CRU (ODE-based), Warpformer (Attention-based), and the requested HyperIMTS.
>
> **1. Empirical Efficiency Comparison**
>
> We report the peak GPU memory usage and average wall-clock time for training and inference (batch size 32).
>
> | Model | Peak Mem (GB) | Training (ms/step) | Inference (ms/step) |
> | :--- | :--- | :--- | :--- |
> | CRU | 0.066 | 429.32 | 163.92 |
> | Warpformer | 13.949 | 289.75 | 113.31 |
> | tPatchGNN | 3.046 | 27.63 | 11.48 |
> | HyperIMTS | 0.459 | 50.05 | 31.14 |
> | **ASTGI (Ours)** | **2.049** | **49.28** | **19.21** |
>
> *   **Speed:** ASTGI achieves a highly favorable trade-off. It is **orders of magnitude faster** (up to $8\times$) than ODE-based (CRU) and complex attention-based (Warpformer) methods. Compared to the graph-based HyperIMTS, ASTGI achieves **comparable training speed and significantly faster inference**, demonstrating that our dynamic graph construction does not introduce a bottleneck.
> *   **Memory:** ASTGI remains lightweight (approx. 2GB), fitting easily onto consumer-grade GPUs, unlike Warpformer which demands heavy resources.
>
> **2. Addressing Computational Complexity in Nearest Neighbor Search**
>
> The reviewer correctly identifies the theoretical complexity bounds. We address this potential bottleneck through a two-tier strategy that leverages both data characteristics and algorithmic optimizations:
>
> 1.  **Efficiency via Sparsity and Vectorization (Standard Regime):** ASTGI operates exclusively on valid observation points, ignoring missing values. In typical IMTS datasets (which are often >80% sparse), the number of effective nodes $N_{obs}$ is significantly smaller than the full grid size ($L \times V$). For these standard settings, we utilize a fully vectorized PyTorch implementation. The reduced $N_{obs}$ combined with the highly optimized dense matrix operations of modern GPUs ensures that the exact top-K search incurs negligible latency compared to the overall forward pass.
> 2.  **Scalable Approximate Search (Large-Scale Regime):** To strictly bound complexity for scenarios where $N$ becomes prohibitively large, our framework integrates the **FAISS** library (as detailed in our code). This allows the model to switch to approximate nearest neighbor search via inverted indexing, reducing the complexity from $O(N^2)$ to **$O(N \log N)$**. This mechanism ensures the model remains computationally feasible and scalable without significant accuracy degradation.

---

### Author Response · Authors · 2025-12-03
**Rebuttal Summary: Rating Increased (4.0->4.7) by the Sole Responsive Reviewer**

**Dear Reviewers, ACs, SACs, and PCs,**

We acknowledge the recent OpenReview technical disruptions and support the remedial actions, confirming our strict adherence to all policies. Crucially, our substantive discussions and **Reviewer FrHL's score increase to 8** were concluded **prior to the incident**.

To assist in the final assessment, we summarize the consensus on our work's **strengths** and the **outcome of the discussion** below:
### Consensus on Strengths
Reviewers agree on the following core contributions:
- **Novel Metric Space Paradigm**: Reviewers recognized our "Spatio-Temporal Point Representation" in a continuous embedding space as a valid and effective approach, distinct from standard interpolation or decay-based methods. (Reviewer LeMp, Reviewer FrHL)
- **Scalability and Efficiency**: The extensive analysis confirmed that our method is computationally efficient and scalable, making it practical for real-world deployment. (Reviewer FrHL, Reviewer 2C1m)
- **Clear Presentation & Reproducibility**: The paper is well-structured, and the code availability supports transparency. (Reviewer 2C1m, Reviewer FrHL)
### Key Actions During Rebuttal
We addressed the reviewers' concerns through:
- **Visual Interpretability (Response to Reviewer 2C1m & LeMp)**: We added **Appendix C (Figure 5)** to visualize the learned causal graphs as explicitly requested by **Reviewer 2C1m**, and **Appendix D (Figure 6)** to visualize the embedding space for **Reviewer LeMp**. These results prove that ASTGI learns meaningful semantic clustering and temporal dependencies.
- **Clarification of Methodological Novelty (Response to Reviewer 2C1m)**: We provided a fundamental distinction between our adaptive metric learning and the static rules used in prior works (HyperIMTS).
- **Efficiency Benchmarking (Response to Reviewer 2C1m & FrHL)**: We reported detailed computational metrics, confirming ASTGI is orders of magnitude faster than ODE-based baselines.
### Reviewer Interaction & Outcome
**Reviewer FrHL** explicitly confirmed concerns were addressed, upgrading their score to **8**. In contrast, **Reviewers 2C1m and LeMp** did not respond.

Despite **Reviewer 2C1m**'s silence, we provided the **Graph Visualization (Fig. 5)** and **Efficiency Analysis** they specifically requested. Regarding their concern on "conceptual overlap with HyperIMTS," we respectfully posit that it stems from an **over-generalization of "node-based modeling."** We have clarified this fundamental misunderstanding as follows:
* **Paradigm Shift (Static vs. Adaptive):** Reviewer 2C1m suggests similarity solely because both methods treat observations as nodes. By this logic, any GNN-based method could be conflated. The critical innovation of ASTGI is **how dependencies are discovered**. HyperIMTS relies on **fixed, predefined heuristics** (specifically, explicit "shared timestamps" to form hyperedges), which forces a rigid structure. In stark contrast, ASTGI maps points into a **learnable continuous metric space**, where topology is dynamically generated via nearest-neighbor search.
* **Independence from Alignment:** Because HyperIMTS relies on shared timestamps, it structurally degenerates on sparse, unaligned data. ASTGI’s metric-space approach is **alignment-free**, enabling direct interaction between asynchronous points based on learned semantic proximity. This is not a trivial variation, but a fundamental methodological departure from rule-based graph construction to fully data-driven metric learning.

We believe this clarification effectively refutes the claim of overlap.

**Table: Reviewers' Scores and Status**
|Reviewer|Interaction Status|Rating|Conf.|Summary of Changes|
|:---|:---|:---|:---|:---|
|**Reviewer FrHL**|**Replied & Increased Score**|**6->8**|4|**Concerns Resolved.** Validated our complexity analysis and accepted the rationale for parameter sharing. Recognized the work as a strong contribution. |
|**Reviewer LeMp**|**No Response to Rebuttal**|4->4|4|**Addressed.** The primary concern was the rationale for Euclidean distance and embedding visualization. We provided the theoretical derivation and added **Appendix D (Figure 6)** to visualize the embedding space, resolving the interpretability concern.|
|**Reviewer 2C1m**|**No Response to Rebuttal**|2->2|3|**Addressed & Clarified.** We provided the requested **Graph Visualization (Appendix C)** and efficiency benchmarks. We also refuted the overlap claim by demonstrating the shift from **fixed heuristics (HyperIMTS)** to **adaptive metric learning**, solving the alignment dependency issue.|
|**Avg.**||**4 -> 4.7**||**Positive trend led by Reviewer FrHL.**|

**Conclusion**
With the **score of 8 from Reviewer FrHL**, the **comprehensive visualizations provided for Reviewers 2C1m and LeMp**, and the **conceptual clarification provided for Reviewer 2C1m**, we believe ASTGI stands as a novel and effective contribution to the field of irregular time series forecasting.

Best regards,
Authors

---

### Meta-Review · Area_Chair_iU6E · 2026-01-02

**Summary:**

This paper presents ASTGI, an adaptive graph-based framework designed for forecasting irregular multivariate time series. It represents discrete observations as nodes in a learnable spatio-temporal domain and captures their interactions via dynamic propagation mechanisms, followed by a query-driven prediction method. Tests on various public benchmarks show that this approach outperforms several leading techniques.

**Reviewer Concerns:**

Reviewer 2C1m raised concerns about substantial conceptual overlap with HyperIMTS (Li et al., ICML 2025) and the lack of qualitative or quantitative analysis of the learned graph structures. After considering the authors’ rebuttal, these concerns are partially addressed, particularly by clarifying the distinctions from HyperIMTS, and the response appears reasonably credible.

Reviewer LeMp noted that the relationship between the two core challenges is insufficiently articulated and that the motivation and contributions of certain modules are not well justified. The authors’ response alleviates some, but not all, of these issues.

Reviewer FrHL expressed concerns about the clarity of the computational complexity analysis, scalability, and the discussion of key hyperparameters. In this case, the rebuttal addresses most of the reviewer’s concerns.

**Reviewer Scores:**

The initial overall scores from reviewers 2C1m, LeMp, and FrHL were 2, 4, and 6, with corresponding confidence scores of 3, 4, and 4. After reviewing the authors’ rebuttal, reviewer 2C1m may reasonably increase the overall score from 2 to 4. Reviewer LeMp could potentially raise the scores for soundness or contribution, and under a more favorable assessment, the overall score may increase from 4 to 6. Reviewer FrHL participated in the discussion prior to the technical disruption and subsequently increased the overall score from 6 to 8.

Overall, the evaluations remain notably divergent. Taking these factors into account, I consider this paper to be borderline acceptable.

---

### Decision · Program_Chairs · 2026-01-26

Accept (Poster)